# An optical neural network using less than 1 photon per multiplication

Tianyu Wang [1✉], Shi-Yuan Ma [1], Logan G. Wright [1,2], Tatsuhiro Onodera [1,2], Brian C. Richard[3] & Peter L. McMahon [1✉]

Deep learning has become a widespread tool in both science and industry. However, continued progress is hampered by the rapid growth in energy costs of ever-larger deep neural networks. Optical neural networks provide a potential means to solve the energy-cost problem faced by deep learning. Here, we experimentally demonstrate an optical neural network based on optical dot products that achieves 99% accuracy on handwritten-digit classification using ~3.1 detected photons per weight multiplication and ~90% accuracy using ~0.66 photons (~2.5 × 10⁻¹⁹ J of optical energy) per weight multiplication. The fundamental principle enabling our sub-photon-per-multiplication demonstration—noise reduction from the accumulation of scalar multiplications in dot-product sums—is applicable to many different optical-neural-network architectures. Our work shows that optical neural networks can achieve accurate results using extremely low optical energies.

[1] School of Applied and Engineering Physics, Cornell University, Ithaca, NY 14853, USA. [2] NTT Physics and Informatics Laboratories, NTT Research, Inc., Sunnyvale, CA 94085, USA. [3] School of Electrical and Computer Engineering, Cornell University, Ithaca, NY 14853, USA. ✉email: tw329@cornell.edu; pmcmahon@cornell.edu

Much of the progress in deep learning over the past decade has been facilitated by the use of deeper and larger models[1], with commensurately larger computation requirements and energy consumption[2]. The growth in energy consumption is unsustainable given the energy efficiency of conventional digital processors[2]. As 80–90% of the cost of large-scale commercial deployments of deep neural networks (DNNs) is due to machine-learning inference[3], there is a strong incentive to develop more energy-efficient hardware that is specialized to DNN inference processing[4].

Optical processors have been proposed as deep-learning accelerators that can in principle achieve better energy efficiency and lower latency than electronic processors[5–9]. For deep learning, optical processors' main proposed role is to implement matrix-vector multiplications[10,11], which are typically the most computationally-intensive operations in DNNs[4].

Theory and simulations have suggested that optical neural networks (ONNs) built using optical matrix-vector multipliers can exhibit extreme energy efficiency surpassing even the fundamental limit of irreversible digital computers[7]. It has been predicted that for sufficiently large vector sizes, matrix-vector multiplication can be performed with an optical energy cost of less than 1 photon ($\sim 10^{-19}$ J) per scalar multiplication, assuming the standard quantum limit for noise[7,8]. This suggests a possibility for optical processors to have an energy advantage of several orders of magnitude over electronic processors using digital multipliers, which currently consume between $10^{-14}$ and $10^{-12}$ J per scalar multiplication[12,13].

The energy efficiency of optical matrix-vector multiplication improves with the sizes of the matrix and vectors that are to be multiplied. With large operands, many constituent scalar multiplication and accumulation operations can be performed in parallel completely in the optical domain, and the costs of conversions between electronic and optical signals can be amortized[8]. Many innovative designs have achieved large-scale parallel processing based on several degrees of freedom unique to optics, including using wavelength multiplexing[14–16], spatial multiplexing in photonic integrated circuits[10,14,16–19], and spatial multiplexing in 3D free-space optical processors[11,20–29]. In many of these architectures, matrix-vector multiplications are performed by computing vector-vector dot products in parallel and concatenating the resulting scalar outputs to form the output vectors.

While ONNs across all multiplexing approaches and architectures are making rapid progress in terms of scaling and energy efficiency[5,15,27–30], the ability of ONNs to operate close to the shot-noise limit of detected photons has not yet been studied in experiments. ONNs can approach the high energy efficiency predicted by theory[7,8] in part by exploiting the parallelism in optical matrix-vector multipliers, and in part because DNNs can be trained to achieve robust performance in the presence of noise[31], and when using low-precision arithmetic operations[32].

Here, we report on our experimental validation of the operation of an ONN in the sub-photon-per-multiplication regime. Based on a piece of customized experimental apparatus performing optical vector-vector dot products, we demonstrate image classification using on average <1 detected photon per scalar multiplication, matching theoretical predictions for the quantum-limited optimal efficiency of ONNs[7].

## Results

**Large-scale optical vector-vector dot products**. To study ONNs in the large-vector limit where accurate sub-photon operation has been predicted to be possible[7,8], we constructed a demonstration setup using the scheme shown in Fig. 1 to compute optical vector-

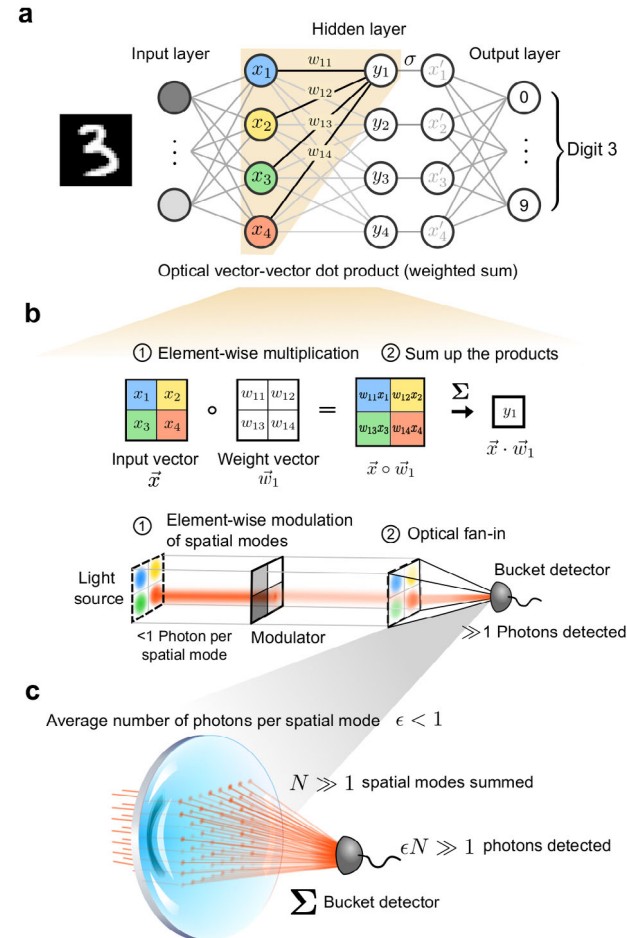

**Fig. 1 An optical vector-vector dot product multiplier for characterizing the optical energy consumption of an ONN. a** The role of optical vector-vector dot products in executing the forward-pass operation in a fully connected neural network. The weighted sums of neural activations are performed optically (shaded area) and the element-wise nonlinear activation functions are performed electronically. Each neuron in the middle (hidden) layer is color-coded to show the correspondence to their representations in (**b**). The shaded area in the neural-network schematic illustrates the neurons and weights involved in one dot product. **b** A step-by-step illustration of the computation of optical vector-vector-dot product between $\vec{\mathbf{x}}$ and $\vec{\mathbf{w}}_\mathbf{1}$. The top row shows mathematically abstract operations, and the bottom row shows the corresponding physical operations with optics. "∘" denotes element-wise multiplication between two vectors of the same size. **c** An illustration of how the optical fan-in operation allows less-than-1-photon-per-scalar multiplication when the vector size is large. A single lens is used to sum the intensities of the spatial modes encoding the element-wise products onto a detector. For sufficiently large vector size $N$, even if each individual spatial mode contains $\epsilon < 1$ photon on average, the total number of photons impinging on the detector $\epsilon N$ will be ≫1.

vector dot products, which can be thought of as a modified Stanford-Vector-Multiplier architecture[21] without optical fan-out (Supplementary Fig. 12). Our experimental scheme and setup are not intended as a prototype to directly compete with conventional electronic processors or even other ONN systems, since it has not been optimized for speed or electrical energy consumption. Instead, it was designed to support a clean and quantitative investigation of the limits of optical energy consumption in large-scale ONNs.

Our setup uses the following scheme to perform vector-vector dot products $y_i = \vec{\mathbf{x}} \bullet \vec{\mathbf{w_i}} = \sum_j x_j w_{ij}$, where $\vec{\mathbf{x}}$ is the input vector of neural activations in the previous layer, and $\vec{\mathbf{w_i}}$ is the weight vector consisting of the weights connecting each neuron in $\vec{\mathbf{x}}$ to the $i$th neuron in the next layer (Fig. 1a). Each element $x_j$ of vector $\vec{\mathbf{x}}$ is encoded in the intensity of a stand-alone spatial mode illuminated by a light source pixel, and each weight $w_{ij}$ is encoded as the transmissivity of a modulator pixel. We used an organic light-emitting diode (OLED) display as the light source and a spatial light modulator (SLM) for intensity modulation. Dot products were computed in two physical steps: (1) Element-wise Multiplication: Each OLED pixel encoding a single vector element $x_j$ was aligned and imaged to a corresponding pixel on the SLM, whose transmissivity was set to be $\propto w_{ij}$. This performs the scalar multiplication $w_{ij}x_j$ (Fig. 1b bottom middle). (2) Optical Fan-in: The intensity-modulated pixels from each block were physically summed by focusing the light transmitted by them onto a detector. The total number of photons impinging on the detector is proportional to the dot product result $y_i$ (Fig. 1b bottom right). The encoding of vector elements in optical intensity constrains the setup to performing dot products with vectors that have non-negative elements, which can be converted to dot products with signed elements[21] (Supplementary Note 11).

Our optical dot product multiplier can perform at most $711 \times 711 = 505,521$ scalar multiplications and additions in parallel, enabled by the alignment of an array of $711 \times 711$ pixels on the OLED display to an array of $711 \times 711$ pixels on the SLM (Supplementary Notes 5–7). The 2D layout of vector $\vec{\mathbf{x}}$ and $\vec{\mathbf{w_i}}$ maximizes the dimension of the vectors by taking full advantage of the 2D imaging plane. Our experimental setup was, with a single pass of light through the setup, capable of performing a single vector-vector dot product with vectors having sizes up to 505,521.

Computing dot products between very large vectors with optical fan-in allows extremely low optical energy consumption, which is a unique advantage offered by optics[33]. For each dot product that the system computes, the summation of the element-wise products is performed by focusing the spatial modes corresponding to the element-wise products onto a single detector. If the vectors have size $N$, then $N$ spatial modes are incoherently summed on the detector. Consequently, the detector's output, which is proportional to the dot-product answer, has a signal-to-noise ratio (SNR) that scales as $\sqrt{N}$ at the shot-noise limit[8]. If the vector size $N$ is sufficiently large, then even if the individual spatial modes each have an average photon number far less than 1, the total number of photons impinging on the detector can be much greater than 1, and so precise readout of the dot-product answer is possible (Fig. 1c).

**Accuracy of sub-photon dot products**. We characterized the accuracy of dot products performed by our setup while varying the number of photons used. In our first characterization experiments, we computed the dot products of randomly chosen pairs of vectors (Fig. 2a; see Methods). The optical signal encoding the dot-product solution was measured by a sensitive photodetector, and the number of photons used for each dot product was controlled by changing the detector integration time and by inserting neutral-density filters immediately after the OLED display (see Methods). The number of photons per multiplication was calculated directly as the total measured optical energy divided by the number of scalar multiplications in the dot product (Supplementary Note 15).

To investigate the possibility of using less than 1 photon per scalar multiplication for large vector sizes, we measured the

numerical precision of dot products between vectors each of size ~0.5 million. With 0.001 photons per scalar multiplication, the error was measured to be ~6% (Fig. 2b; see Supplementary Note 12 for the details of RMS-error calculation); the dominant contribution to this error was from shot noise at the detector (Supplementary Note 8). As we increased the number of photons used, the error decreased until it reached a minimum of ~0.2% at 2 photons per multiplication or higher (Fig. 2b). We hypothesize that the dominant sources of error at high photon counts are imperfect imaging of the OLED display pixels to SLM pixels, and crosstalk between SLM pixels. To enable comparison between the experimentally achieved analog numerical precision with the numerical precision in digital processors, we can interpret each measured analog error percentage (Fig. 2b) as corresponding to an effective bit-precision for the computed dot product's answer. Using the metric of noise-equivalent bits[8], an analog RMS error of 6% corresponds to 4 bits, and 0.2% RMS error corresponds to ~9 bits (see Methods).

We also verified that we could compute dot products between shorter vectors when using low numbers of photons per scalar multiplication (Fig. 2c). For photon budgets ranging from 0.001 to 0.1 photons per multiplication, the numerical error was dominated by shot noise for all vector sizes tested. When the number of photons used was sufficiently large, the error was no longer dominated by shot noise, which is consistent with the single-vector-size results shown in Fig. 2b. For every photon budget tested, dot products between larger vectors had lower error; we attribute this to dot products between larger vectors involving the effective averaging of larger numbers of terms.

**ONN using sub-photon multiplications**. Having characterized the accuracy of our experimental setup for performing multiplication operations with random vectors, we set out to demonstrate its use as the core of an experimental ONN implementation. We realized an ONN comprised of fully connected layers where the vector-vector dot products between each layer were computed optically using our experimental setup, and where the digital biases and nonlinearity were applied electronically (using a digital processor) between each layer.

Our main goal was to determine the extent to which our ONN could tolerate multiplication inaccuracy resulting from the use of a very limited photon budget. Theoretical studies indicate DNNs can be constructed and trained to be intrinsically resilient to noise, including photon shot noise and technical noise[7,31]. Our approach was to run a trained neural network with our setup and measure the classification accuracy as a function of the number of photons used. We used handwritten-digit classification with the MNIST dataset as our benchmark task and trained a 4-layer fully connected multi-layer perceptron (MLP) (Fig. 3a) with a back-propagation procedure designed for use with low-precision inference hardware (Quantization-Aware Training—QAT[34]; see Methods).

We evaluated the first 130 test images in the MNIST dataset under 5 different photon budgets: 0.03, 0.16, 0.32, 0.66, and 3.1 average detected photons per scalar multiplication (Fig. 3b, center panel, orange dots; Supplementary Note 15 describes the procedure for calculating photons per scalar multiplication). We found that using 3.1 photons per multiplication led to a classification accuracy of ~99% (Fig. 3b, top-right panel), which was almost identical to the accuracy (99%) of the same trained neural network run on a digital computer without any noise. In the sub-photon regime, using 0.66 photons per multiplication, the ONN achieved ~90% classification accuracy (Fig. 3b, top-middle panel). The reported experimental accuracies were obtained with single-shot execution of the neural network without any

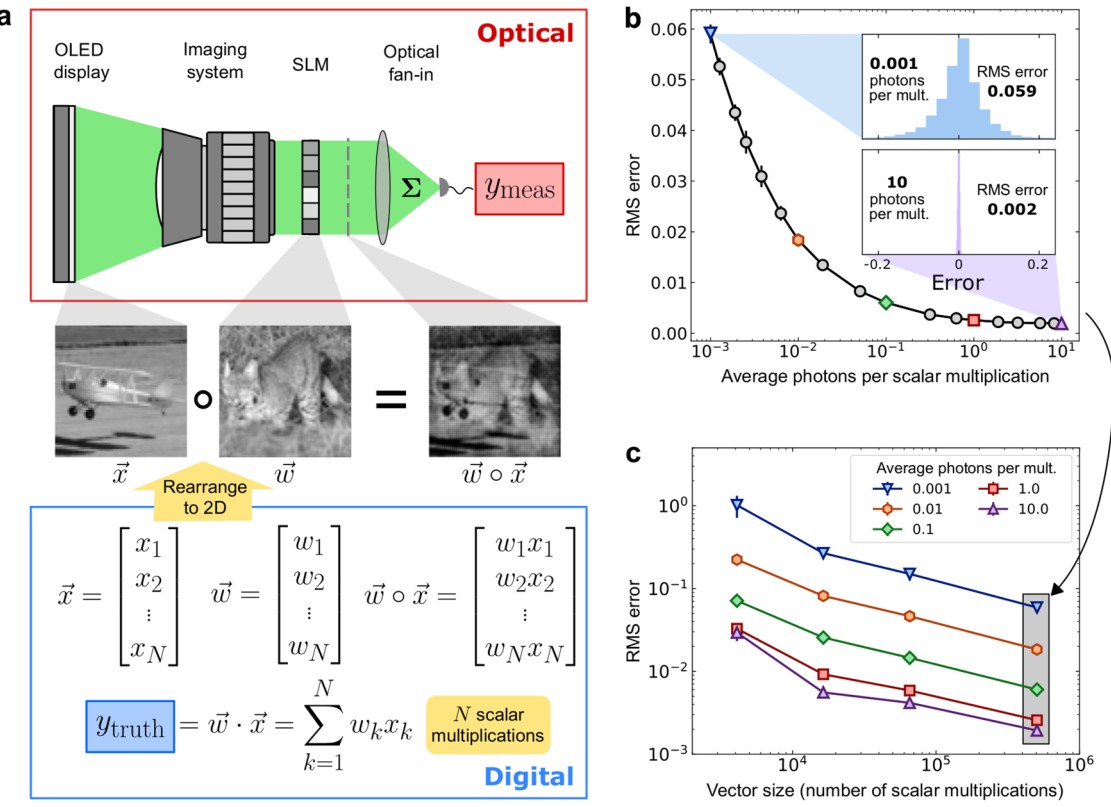

**Fig. 2 Vector-vector dot products were computed with high accuracy using as few as 0.001 photons per scalar multiplication. a** The procedure for characterizing optical vector-vector dot products. $N$-pixel images were used as test vectors by interpreting each image as an $N$-dimensional vector. The setup was used to compute the dot products between many different random pairs of vectors, with each computation producing a result $y_{meas}$ (top and center rows; example experimental measurement of element-wise multiplication $\vec{w} \circ \vec{x}$ was captured with a camera before optical fan-in for illustrative purposes). The dot-product ground truth $y_{truth}$ was computed on a digital computer (bottom row). The error was calculated as $y_{meas} - y_{truth}$. OLED: organic light-emitting display; SLM: spatial light modulator. **b** The root-mean-square (RMS) error of the dot product computation as a function of the average number of detected photons per scalar multiplication. The vector length $N$ was ~0.5 million (711 × 711). The error bars show 10× the standard deviation of the RMS error, calculated using repeated measurements. The insets show error histograms (over different vector pairs and repeated measurements) from experiments using 10 and 0.001 photons per multiplication, respectively. **c** The RMS error as a function of the vector size $N$. For each vector size, the RMS error was computed using five different photon budgets, ranging from 0.001 to 10 photons per scalar multiplication. The shaded column indicates data points that are also shown in (**b**). The error bars show 3× the standard deviation of the RMS error, calculated using repeated measurements.

repetition. The simulation curve (Fig. 3b, center panel, dark-blue line) was produced by executing the same trained neural network model completely on a digital computer with simulated shot noise, but without modeling any other physical process in the experimental setup. The only cause for the classification accuracy of the simulation curve to drop at lower photon budgets is the presence of the simulated photon shot noise. The reasonable agreement between the experimental and the simulation results supports the hypothesis that the classification accuracy of the experimental ONN was primarily limited by the photon detection shot noise at low photon budgets, and indeed the SNR measured by the detector was close to the shot-noise limit (Supplementary Fig. 10).

To achieve an accuracy of 99%, the detected optical energy per inference of a handwritten digit was ~107 fJ (Fig. 3b). For the weight matrices used in these experiments, the average SLM transmission was ~46%, so when considering the unavoidable loss at the SLM, the total optical energy needed for each inference was ~230 fJ. For comparison, this energy is less than the energy typically used for only a single float-point scalar multiplication in electronic processors[35], and our model required 90,384 scalar multiplications per inference (Supplementary Note 15) with each optical operation simply replacing a corresponding operation in

the digital version of the same fully trained neural network (see the concept of isomorphism in ONNs in Ref. [5]).

## Discussion

Here we have presented experimental results that support the notion that optical neural networks can in principle[5–8] have a fundamental energy advantage over electronic neural-network implementations. We showed that ONNs can operate in a photon-budget regime in which the standard quantum limit (i.e., optical shot noise) governs the achievable accuracy. In particular, we achieved high classification accuracies using our ONN even when restricted to a photon budget of less than one detected photon per scalar multiplication.

The sub-photon-per-multiplication result in this study can be interpreted in the following way: in a single pass of light through the experimental setup for computing element-wise products (i.e., past the modulator in Fig. 1b, but before optical fan-in), the number of measured photons in each spatial mode would be an integer if each spatial mode were measured separately. However, after optical fan-in, only the total number of photons across all the spatial modes is measured. Since many of the spatial modes are likely to contribute zero measured photon when they have low

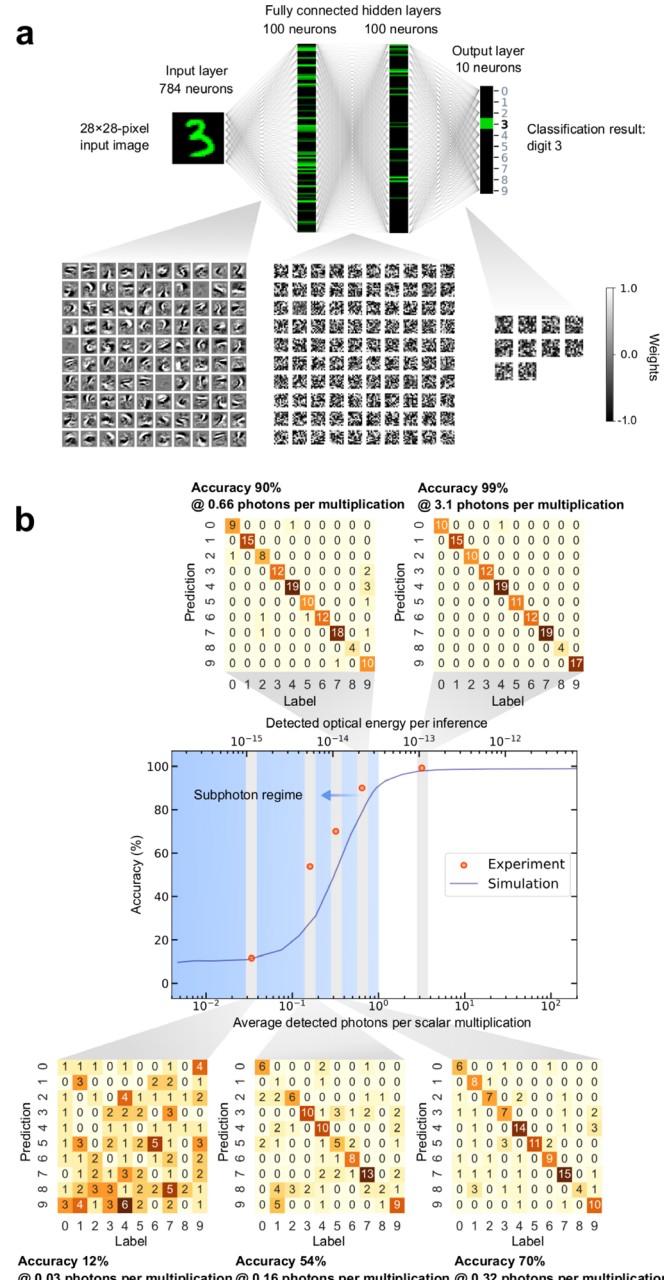

**Fig. 3 MNIST handwritten-digit classification was demonstrated with an optical neural network using less than one detected photon per scalar multiplication.** **a** Illustration of the 4-layer neural network for handwritten-digit classification that we implemented as an ONN. Top panel: the neural network is composed of a sequence of fully connected layers represented as either a block (input image) or vertical bar (hidden and output layers) comprising green pixels, the brightness of which is proportional to the activation of each neuron. The weights of the connections between neurons for all four layers are visualized; the pixel values in each square array (bottom panel) indicate the weights from all the neurons in one layer to one of the neurons in the next layer. **b** Classification accuracy tested using the MNIST dataset as a function of optical energy consumption (middle panel), and confusion matrices of each corresponding experiment data point (top and bottom panels). The detected optical energy per inference is defined as the total optical energy received by the photodetector during the execution of the three matrix-vector multiplications comprising a single forward pass through the entire neural network. The shaded area in the plot indicates the regime where less than 1 detected photon was used for each scalar multiplication. Each shaded vertical bar in the plot directs to the confusion matrix corresponding to each experimental data point.

expected photon number, the number of photons *averaged* across all the spatial modes, each performing one scalar multiplication in our ONN, can be between 0 and 1.

Our results indicate that ONNs not only have the potential to achieve orders of magnitude higher energy efficiency than electronic processors on the per-operation basis, but even more so on the per-inference basis. In the broader DNN accelerator community, the energy-per-inference metric is a standard means of quantifying system efficiency[36]. The use of this metric is motivated by the fact that it is uniformly defined across different physical platforms, while the definition and efficacy of an operation varies depending on the implementation details of a particular processor, especially for analog computing. Here we report ~99% accuracy achieved on MNIST handwritten-digit classification with at least ~230 fJ total optical energy required to perform all the matrix-vector multiplications for each inference. We now briefly consider what these results suggest for the future potential of ONNs. Several hardware analyses[30,37–39] have

suggested that optical energy consumption could account for 1% of the system's total (i.e., of the sum of the electrical and optical) energy consumption; such a foreseeable ONN system would consume on the order of $10^{-11}$ J per inference. For comparison, running the same neural-network model that we obtained our results with (Fig. 3a) on state-of-the-art electronic DNN accelerators[12] would require $10^{-13}$ J/operation × 90,384 operations ~$10^{-8}$ J per inference (our experiment allows such a direct comparison since the model run on our optical setup was the same as the one run on a digital computer). For additional comparison, some custom electronic chips report whole-system energy efficiency on the order of $10^{-7}$ J per inference for MNIST classification with >97% accuracy[40,41]. The achievable energy advantage of ONNs over electronic processors would likely vary with the dataset and neural-network structure. Nevertheless, our work provides a starting point for the experimental determination of the required optical energy in ONNs for different machine-learning tasks and neural-network architectures[19,23,24,26,42].

Our proof-of-principle results for sub-photon-per-multiplication ONN operation are likely to translate to other ONN platforms. Many existing ONN schemes can be boiled down to parallel execution of vector-vector dot products by summing element-wise-modulated spatial[20–24], temporal[7], or frequency modes[14–16]. When summing incoherent spatial modes on a detector, the measured energy equals the sum of energy of each mode. The same statement applies to summing optical modes of different wavelengths[14–16] or light pulses arriving at different times[7] on the same detector, even if coherent light is used in these cases. For this reason, while we chose to use spatial modes to demonstrate the computation of vector-vector dot products, our results should apply generally to schemes that involve the summation of a sufficiently large number of optical modes (be they spatial, frequency, or temporal modes) at each detector[7,8,39].

The focus on the required optical energy in ONNs instead of the whole-system energy in this paper is motivated by the fact that the former is approximately universal across different platforms and implementations, while the latter is more specific to a particular hardware implementation. Although the data throughput rate in our setup was limited by the update rate of input hardware (10 s Hz for both OLED and SLM), the detector was fast enough (with ~100 ns minimum integration time) to allow the extension of our conclusions based on photon detection to high-speed systems. Using our results on optical-energy

consumption, the best-case whole-system energy consumption of various architectures can be estimated based on several detailed studies focused on hardware analysis[7,8,16,28,37,39].

One critical step towards building practical ONNs with high overall energy efficiency is to design a full-scale optical matrix-vector multiplier with optical fan-out and fan-in (Supplementary Notes 9 and 10), integrated with fast and highly efficient modulators[43] and detectors[44]. While the 2D-block matrix-vector multiplier used in this work is not the architecture most closely matched to incorporating integrated-photonics modules in the short term, it may serve as a viable platform for image-processing tasks involving incoherent light sources, which are common in biomedical imaging and robotics[45].

More broadly, the ability to perform matrix-vector multiplications more efficiently could find applications beyond neural networks, including in other machine-learning algorithms[46] and for combinatorial-optimization heuristics[47–49].

## Methods

**Experimental setup.** We used the OLED display of an Android phone (Google Pixel 2016) as the incoherent light source for encoding input vectors in our experimental setup. Only green pixels (with an emission spectrum centered around 525 nm) were used in the experiments; the OLED display contains an array of ~2 million (1920 × 1080) green pixels that can be refreshed at 60 Hz at most (Supplementary Note 2). Custom Android software was developed to load bitmap images onto the OLED display through Python scripts running on a control computer. The phone was found capable of displaying 124 distinct brightness levels (~7 bits) in a linear brightness ramp (Supplementary Note 2). At the beginning of each matrix-vector-multiplication computation, the vector was reshaped into a 2D block (Fig. 1a) and displayed as an image on the phone screen for the duration of the computation. The brightness of each OLED pixel was set to be proportional to the value of the non-negative vector element it encoded. Fan-out of the vector elements was performed by duplicating the vector block on the OLED display.

Scalar multiplication of vector elements with non-negative numbers was performed by intensity modulation of the light that was emitted from the OLED pixels. An intensity-modulation module was implemented by combining a phase-only reflective liquid-crystal spatial light modulator (SLM, P1920-500-1100-HDMI, Meadowlark) with a polarizing beam splitter and a half-wave plate in a double-pass configuration (Supplementary Note 3). An intensity look-up table (LUT) was created to map SLM pixel values to transmission percentages, with an 8-bit resolution (Supplementary Note 3).

Element-wise multiplication between two vectors $\vec{w}$ and $\vec{x}$ was performed by aligning the image of each OLED pixel (encoding an element of $\vec{x}$) to its counterpart pixel on the SLM (encoding an element of $\vec{w}$) (Fig. 1b). By implementing such pixel-to-pixel alignment, as opposed to aligning patches of pixels to patches of pixels, we maximized the size of the vector-vector multiplication that could be performed by this setup. A zoom-lens system (Resolve 4 K, Navitar) was employed to de-magnify the image of the OLED pixels by ~0.16× to match the pixel pitch of the SLM (Supplementary Note 5). The image of each OLED pixel was diffraction-limited with a spot diameter of ~6.5 μm, which is smaller than the 9.2 μm size of pixels in the SLM, to avoid crosstalk between neighboring pixels. Pixel-to-pixel alignment was achieved for ~0.5 million pixels (Supplementary Note 5). This enabled the setup to perform vector-vector dot products with 0.5-million-dimensional vectors in single passes of light through the setup (Fig. 2b). The optical fan-in operation was performed by focusing the modulated light field onto a detector, through a $4f$ system consisting of the rear adapter of the zoom-lens system and an objective lens (XLFLUOR4x/340, NA = 0.28, Olympus) (Supplementary Fig. 1 and Supplementary Note 9).

The detector measured optical power by integrating the photon flux impinging on the detector's active area over a specified time window. Different types of detector were employed for different experiments. A multi-pixel photon counter (MPPC, C13366-3050GA, Hamamatsu) was used as a bucket detector for low-light-level measurements. This detector has a large dynamic range (pW to nW) and moderately high bandwidth (~3 MHz) (Supplementary Note 4). The MPPC outputted a single voltage signal representing the integrated optical energy of the spatial modes focused onto the detector area by the optical fan-in operation (Supplementary Note 9). The MPPC is capable of resolving the arrival time of single-photon events for low photon fluxes (<$10^6$ per second); for higher fluxes that exceed the bandwidth of MPPC (~3 MHz), the MPPC output voltage is proportional to the instantaneous optical power (Supplementary Fig. 6 and Supplementary Note 4). The SNR of the measurements made with the MPPC was roughly half of the SNR expected for a shot-noise-limited measurement (Supplementary Note 8). The integration time of the MPPC was set between 100 ns and 1 ms for the experiments shown in Fig. 2, and between 1 μs to 60 μs for the experiments shown in Fig. 3. Since the MPPC does not provide a spatial resolution within its active area, it effectively acts as a single-pixel detector (Fig. 1c) and

consequently could only be used to read out one dot product at a time. For the parallel computation of multiple dot products (as is desirable when performing matrix-vector multiplications that are decomposed into many vector-vector dot products), a CMOS camera (Moment-95B, monochromatic, Teledyne) was used. The intensity of the modulated light field was captured by the camera as an image, which was divided into regions of interest (ROIs), each representing the result of an element-wise product of two vectors. The pixels in each ROI could be then summed digitally to obtain the total photon counts, which correspond to the value of the dot product between the two vectors. Compared to the MPPC, the CMOS camera was able to capture the spatial distribution of the modulated light but could not be used for the low-photon-budget experiments due to its much higher readout noise (~2 electrons per pixel) and long frame-exposure time (≥10 μs). Consequently, the camera was only used for setup alignment and for visualizing the element-wise products of two vectors with highoptical powers, and the MPPC was used for the principal experiments in this work—vector-vector dot-product calculation and machine-learning inference with an ONN involving low numbers of photons per scalar multiplication (Fig. 2 and Fig. 3).

**Evaluation of dot-product accuracy.** The numerical accuracy of dot products was characterized with pairs of vectors consisting of non-negative elements; since there is a straightforward procedural modification to handle vectors whose elements are signed numbers, the results obtained are general (Supplementary Note 11). The dot-product answers were normalized such that the answers for all the vector pairs used to fall between 0 and 1; this normalization was performed such that the difference between true and measured answers could be interpreted as the achievable accuracy in comparison to the full dynamic range of all possible answers (for the equations used for the error calculation, see Supplementary Note 12).

Before the accuracy-characterization experiments were performed, the setup was calibrated by recording the output of the detector for many different pairs of input vectors and fitting the linear relationship between the ground truth of the dot-product answer and the detector's output (Supplementary Note 12).

The vector pairs used for accuracy characterization were generated from randomly chosen grayscale natural-scene images (STL-10 dataset[50]). The error of each computed dot product was defined as the difference between the measured dot-product result and the ground truth calculated by a digital computer (Fig. 2a). The number of photons detected for each dot product was tuned by controlling the integration time window of the detector (Supplementary Note 12). The measurements were repeated many times to capture the error distribution resulting from noise. For each vector size displayed in Fig. 2c, the dot products for 100 vector pairs were computed. The root-mean-square (RMS) error was calculated based on data collected for different vector pairs and multiple measurement trials. Therefore, the RMS error includes contributions from both the systematic error and trial-to-trial error resulting from noise. The RMS error can be interpreted as the "expected" error from a single-shot computation of a dot product with the setup. The noise equivalent bits were calculated using the formula[8] NEB = $-\log_2$(RMS Error).

**Training of noise-resilient neural networks.** To perform handwritten-digit classification, we trained a neural network with 4 fully connected layers (Fig. 3a). The input layer consists of 784 neurons, corresponding to the 28 × 28 = 784 pixels in grayscale images of handwritten digits. This is followed by two fully connected hidden layers with 100 neurons each. We used ReLU[51] as the nonlinear activation function. The output layer has 10 neurons; each neuron corresponds to a digit from 0 to 9, and the prediction of which digit is contained in the input image is made based on which of the output neurons has the largest value. The neural network was implemented and trained in PyTorch[52]. The training of the neural network was conducted exclusively on a digital computer (our optical experiments perform neural-network inference only). To improve the robustness of the model against numerical error, we employed quantization-aware training (QAT)[34], which was set to quantize the activations of neurons to 4 bits and weights to 5 bits. The PyTorch implementation of QAT was adapted from Ref. [53]. In addition, we performed data augmentation: we applied small random affine transformations and convolutions to the input images during training. This is a standard technique in neural-network training for image-classification tasks to avoid overfitting[34] and intuitively should also improve the model's tolerance to potential hardware imperfections (e.g., image distortion and blurring). The training parameters we used are documented in Supplementary Note 13. The training methods used not only effectively improved model robustness against numerical errors but also helped to reduce the optical energy consumption during inference. We note that the 4-bit quantization of neuron activations was only performed during training, and not during the inference experiments conducted with the optical setup: the activations were loaded onto the OLED display using the full available precision (7 bits).

**Optical neural networks with controlled photon budgets.** To execute the trained neural network with the optical vector-vector dot product multiplier, we needed to perform 3 different matrix-vector multiplications, each responsible for the forward propagation from one layer to the next. The weights of each matrix of the MLP model were loaded onto the SLM, and the vector encoding the neuron values for a particular layer was loaded onto the OLED display. (There is a technicality associated with the handling of negative values, as was mentioned in the above

Methods section on dot-product characterization and is explained in detail in Supplementary Note 11). We performed matrix-vector multiplication as a set of vector-vector dot products. For each vector-vector dot product, the total photon counts (or optical energy) measured by the detector were mapped to the answer of the dot product through a predetermined calibration curve. The calibration curve was made using the first 10 samples of the MNIST test dataset by fitting the measured photon counts to the ground truth of the dot products (Supplementary Note 14). The number of photons per multiplication was controlled by adjusting the detector's integration time (Supplementary Note 12). The measured dot-product results were communicated to a digital computer where bias terms were added and the nonlinear activation function (ReLU) was applied. The resulting neuron activations of each hidden layer were used as the input vector to the matrix-vector multiplication for the next weight matrix. At the output layer, the prediction was made in a digital computer based on the neuron with the highest value.

## Data availability

The raw data generated in this study, including the characterization of dot-product accuracy (Fig. 2) and the layer-by-layer execution of the ONN (Fig. 3), along with the code used to analyze them, have been deposited in the Zenodo database under the permanent link: https://doi.org/10.5281/zenodo.4722066.

## Code availability

The code we used to train neural networks with QAT in PyTorch and controlling the experimental devices is available at: https://doi.org/10.5281/zenodo.4722066.

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

## Acknowledgements

P.L.M. acknowledges membership of the CIFAR Quantum Information Science Program as an Azrieli Global Scholar. The authors wish to thank the following funding agencies for financial supports: NTT Research: T.W., S.-Y. M., L.G.W., T.O. and P.L.M.; Cornell Neurotech the Mong Fellow Program: T.W. and L.G.W.; NSF (award CCF-1918549): P.L.M. We thank Frank Wise for the loan of spatial light modulators, and Chris Xu for the loan of imaging lenses that were used in our preliminary experiments. We thank Ryan Hamerly for a helpful discussion on the energy scaling of optical fan-in. We thank Alen Senanian for sharing equipment and controlling code. We thank Irena Hwang, Anna Clemens, and Francis Chen for editing the paper.

## Author contributions

T.W., L.G.W. and P.L.M. conceived the project. T.W., S.-Y.M. and L.G.W. designed the experiments. T.W. and S.-Y.M. performed the experiments. B.C.R. developed the display control software. T.W., T.O., and L.G.W. contributed to the neural-network training. T.W. and S.-Y.M. analyzed the data. T.W., S.-Y.M. and P.L.M. prepared the paper, with revisions from L.G.W. and T.O.; P.L.M. supervised the project.

## Competing interests

T.W. and P.L.M. are listed as inventors on a U.S. provisional patent application (No. 63/149,974) on the techniques to implement 2D-block optical vector-vector dot product multipliers and matrix-vector multipliers.
