## [Peer Review File · Nature Communications]

An optical neural network using less than 1 photon per multiplicationEditorial Note: This manuscript has been previously reviewed at another journal that is not operating a transparent peer review scheme. This document only contains reviewer comments and rebuttal letters for versions considered at *Nature Communications*.

REVIEWERS' COMMENTS

Reviewer #1 (Remarks to the Author):

I appreciate the authors' comprehensive and accurate responses. All my concerns have been sufficiently addressed, and I do not have any further comments. I believe this work, proving the ultimate potential of optical neural networks at sub-photon's energy consumption, will bring about significant advances to this field. As such, I support the publication of this work.

Reviewer #2 (Remarks to the Author):

The authors have addressed my many comments and suggestions in depth and in a satisfactory manner. The message is also generally now clearer, focusing on the key (and important) novelty of the paper which is the low number of photons per ops.

As I was saying in my previous review, I believe the paper is an excellent match for Nature Communications, and I support publication.

We thank all three referees for their comments and suggestions. Several of the suggestions have helped us to revise the manuscript significantly: we now have placed more focus on the core scientific message of this study and de-emphasized some of the technical points that did not help to support the central message of the paper.

Before addressing each referee's comments point-by-point, we would first like to summarize the main changes that have been made to the manuscript and supplementary materials.

Summary of changes:

1. In the revised main text, we clarify that the primary role of our experimental setup is to serve as scientific apparatus rather than a complete engineering solution for competitive performance, or a showcase of some radical new design concepts. More specifically, our setup was designed to study optical vector-vector dot products with large vector sizes. Such an architecture choice enabled us to systematically investigate the limit of the energy efficiency of optical neural networks operating at an ultra-low-optical-energy regime governed by detected photon shot noise.
2. To be more transparent about the performance of our experimental setup and measurement procedures, we have added a new section in supplementary materials to tally optical and electronic operations during each execution of the optical neural network, together with a detailed calculation of optical energy consumption and efficiency, as well as the whole-system energy consumption and efficiency.
3. We have made changes in the discussion section to:
 - a. Interpret the physical meaning of the experimental result where on average less than one photon was detected to compute each multiplication.
 - b. Discuss why our conclusion is not restricted to our specific experimental scheme or free-space optical neural networks but applies more generally to optical neural networks of different architecture across different platforms.
 - c. Explain that the optical energy consumption demonstrated in this work, either in terms of per operation or per inference, presents convincing evidence that optical processors, regardless of architecture or platform, have the potential to achieve orders of magnitude higher energy efficiency than current digital electronic processors. Even though such a conclusion is not based on actually implementing a full system achieving this performance, it is based on careful measurement of optical energy and reasonable estimation of the whole-system energy efficiency within the range of currently available engineering capacities.

In addition to having revised manuscript and supplementary material, we have highlighted the major changes in the revised manuscript in response to the referees' comments below.

Referee #1

The paper by Wang et al. reports a diffractive optical neural network using less than 1 photon per multiplication. The demonstrated ultra-high energy efficiency is exciting and significant, revealing the potentials of optical neuromorphic computing hardware. However, I have several concerns such that I cannot support the publication of this work in Nature, at least in its current form.

We appreciate the referee's thoughtful comments and appreciation that our experimental results "reveal the potential of optical neuromorphic computing hardware". We have made changes to the manuscript to more clearly explain what this potential is and how the main scientific message of this work is different from prior works. Meanwhile, we also clarify in the manuscript that our setup was designed to enable us to study the optical energy efficiency limit of optical neural networks (see change 1), instead of as a showcase of some radical new concepts for free-space-optical processor designs, or to compete with prior works in terms of engineering metrics (e.g., data throughput rate or whole-system energy efficiency).

Below, we address the specific concerns raised by the referee.

R1.1

The energy efficiency of the optical neural network should not be determined only by the number of photons impinging the detector, it should be determined by--"the sum of power consumption of all used equipment (as calculated in [29], Supplementary Note 2)" / "the computing speed", where the computing speed is related to the throughput (the data write/read rate of the OLED and the detector) and the parallelism of the system (505,521). As such, I think the energy efficiency of this system is much higher than "1 photon per multiplication". The energy efficiency should be calculated following [29] and compared with this prior work, to demonstrate the advance of this work.

R1.1

We agree with the referee that, from the perspective of constructing an optical neural network that delivers a benefit to an engineering customer, the energy efficiency of an optical processor should be measured by the whole-system energy instead of just the detected optical energy – this is ultimately the metric that matters. In the revised manuscript, we have made the important distinction between the whole-system and optical energy efficiency, and also have included a new Supplementary Note 15 to show the whole-system energy consumption and efficiency. It is true that the whole-system energy consumption spent on each operation was much larger than the energy of one photon, when non-optical energy was included.

However, we have rewritten parts of the manuscript to emphasize that the primary goal of this study is to provide convincing evidence that optical neural networks can operate in an ultra-energy-efficient regime governed by detection noise, which was only predicted by theory [7,8] before. This provides experimental evidence on how noise-resilient and energy-efficient optical neural networks can be in principle. The measurement of detected optical energy fully supports

our main conclusion (of experiments agreeing with the theoretical predictions that optical neural networks can operate with high accuracy even with <1 photon per weight multiplication), and it is more meaningful to analyze our experimental results with a focus on the optical energy rather than the whole-system energy consumption alone, for the following reasons:

1). In this study, we made an experimental investigation on how detected photon shot noise (plus some amount of inevitable device noise) fundamentally limits the energy efficiency of optical neural networks. Under this physical noise limit, the sub-photon-per-multiplication regime had been previously analyzed in theory [7, 8] and was first systematically studied experimentally in our work. The detected optical energy was reported since photon shot noise is determined by the number of detected photons. In this view, we note that the system-level energy consumption in the unit of J/operation includes non-optical energy consumption as well; it alone does not provide enough information to indicate how close the optical neural network is operating near this noise limit.

2). Energy efficiency reported in optical energy consumption provides a means to estimate the best possible whole-system energy efficiency of other optical neural networks, even with different designs. As explained in change 3b, we intended to show that optical neural networks, across different engineering platforms (free-space or integrated photonics) or employing different multiplexing methods (e.g., spatial, temporal, and wavelength multiplexing), all have the potential to achieve the level of noise resilience and optical energy efficiency demonstrated in this work. Our conclusion is generalizable because the direct mapping from digital vector-vector dot products to optical dot products, is employed in many different designs of optical neural networks (e.g., Refs. [7, 14, 15, 19, 21, 22, 27]) as well as in this work. Using the optical energy consumption reported in this work, in combination with design-specific engineering parameters (such as laser efficiency, waveguide and modulator loss etc.), one can estimate the best-scenario whole-system energy consumption of an optical neural network of different system architecture.

Changes to Supplementary Materials:

- We have supplied the whole system energy consumption and efficiency as experiment conditions in Supplementary Note 15.

R1.2

Some of the claims of previous research works are not comprehensive or accurate enough. Page 3, Line 75: *“To date, across all multiplexing approaches and architectures, demonstrations of analog ONNs have involved small vector-vector dot products (as a fundamental operation in implementing convolutional layers14,15 and fully connected layers27) or matrix-vector multiplications (for realizing fully connected layers10): the vectors30 have been limited to sizes of at most 64.”* --- the vector size reaches: 250,000 in [15]; 200×200 in [11]; 560×560 in [29], etc ...

As such, the following claim is not convincing:

“This is substantially below the scale (vector sizes $>10^3$) at which sub-photon-per-multiplication

energy efficiency is predicted to be feasible. This is the fundamental reason that the optical energy consumption in recently demonstrated optical processors is still several orders of magnitude higher than that of theoretical predictions.”

“While the potential for large-scale operation exists based on the available parallelism in free-space spatial modes, this potential has not yet been realized.”

In the initial version of our manuscript, we used rather strict criteria to define which work we were considering for comparison, and more importantly, we didn't explain what those criteria were or how we chose them. Based on the referee's comment, we can see how this could cause confusion. Since different optical neural networks implement the linear layers of artificial neural networks with different approaches, it is not always easy to compare them by focusing only on one parameter without context. When we were making the quoted statements, we intended to only compare the vector size of our setup to devices that perform exactly the same kind of operation (i.e., optical matrix-vector multiplication through parallel dot products, each with an arbitrary weight vector specified by the corresponding weights in a trained digital neural network). Through the referee's comments, we realized the statements we made could cause unintended confusion to readers and were non-essential for supporting the central message of this paper. For these reasons, we have removed these statements about the previous works.

Changes:

- We have removed the statements mentioned by the referee from the manuscript.

R1.3

I agree that the energy efficiency of free-space optical neural networks relies on the number of operands performed optically to amortize the costs of OE/EO conversions. However, the number of optical operands in [11] seems to be much larger than this work.

--[11] Supplementary: “This corresponds to $200 \times 200 \times 5 = 0.2$ million neurons (each containing a trainable phase term) and $(200 \times 200) \times 2 \times 5 = 8.0$ billion connections (including the connections to the output layer).”

--This work: 505,521 pixels, equivalent to 505,521 (0.5 million) connections.

R1.3

In comparison to Ref. [11], this work aimed at a quite different aspect of optical-neural-network energy efficiency. Even though both schemes are categorized as free-space optical neural networks and are able to compute optical dot products with large vector sizes, there are still some significant differences caused by different experimental requirements. Based on this understanding, here we tentatively make the following technical comparison to illustrate the difference between the operations performed by the two schemes:

1. The main purpose of our experimental apparatus is to allow an exact optical implementation of a standard neural network model trained to run on digital computers (a pre-trained multilayer perceptron). In contrast, the device in Ref. [11] uses a model that, while potentially more suitable to scalable hardware, cannot be directly compared

operation-by-operation to standard neural networks, making it challenging to directly compare its per-operation performance to neural networks deployed on digital processors.

2. The setup in Ref. [11] relies on free-space optical diffraction with phase modulation to realize matrix-vector multiplication. Our setup performs vector-vector dot products through optical imaging and light intensity modulation. The behavior of the system is largely captured by ray optics without using much diffractive properties of light. Therefore, the experimental setup in this study should be better understood as a modified Stanford Matrix-vector Multiplier (in subsection “Large-scale optical vector-vector dot products” and Supplementary Note 10) instead of being closely related to diffractive neural networks.
3. The setup in Ref. [11] performs matrix-vector multiplication for each flight of light through the system. The vector size is $200 \times 200 = 40,000$, and the matrix elements are encoded by 5 diffractive phase plates in sequence. Since each phase plate also has $200 \times 200 = 40,000$ pixels (neurons), for a total of five layers, the total number of independently programmable parameters for connection weights is $(40k \times 2) \times 5 = 400k$ ($\times 2$ indicates both amplitude and phase information). In comparison, our setup performs a single dot product for each flight of light through the system, the vector size is at most $\sim 500k$, but all the $\sim 500k$ weights are independent and can be set to arbitrary non-negative numbers (also can be extended to signed numbers with some overhead). For these reasons, this work performs a very different operation from the setup in Ref. [11]: even though the number of operations or neuron-to-neuron connections is lower than in Ref. [11] (0.5 million in this work vs. 8 billion [11]), the number of tunable parameters (0.5 million in this work vs. 0.4 million [11]) and vector sizes (0.5 million in this work vs. 0.04 million [11]) are not lower. The independently tunable matrix (neuron-to-neuron connection) weights shown in this work are critical for our demonstration, which involves one-to-one mapping of the operations in a digital neural network directly to the optical setup.

As each neural connection linearly corresponds to the number of optical operands, [11] seems to have a much higher potential of achieving ultra-high energy efficiencies. This leads me to a question--does this work achieve: the highest energy efficiency so far among optical neural networks, or first point out optics' advantage in energy efficiency? Both achievements are decent, except that the latter is not significant enough to be published in Nature.

We agree with the referee that the aforementioned optical neural networks [11, 15, 29] can perform optical vector-vector dot products with very large vector sizes and have the potential to achieve a similar level of optical energy efficiency as shown in this work. In addition, these works have demonstrated novel optical-neural-network design concepts with competitive whole-system energy efficiency and computational speed. Compared to these prior works, our work still makes the following original contribution to the field of optical neuromorphic computing: we conducted the first systematic investigation on the accuracy of optical dot products and the classification accuracy of optical neural networks near the physical noise limit imposed by photon detection, which verified earlier theoretical predictions [7, 8]. Therefore, our work still presents important and novel experimental findings that we believe are of great reference value

to the general community of optical neuromorphic computing.

As such, comparisons (fan-in, number of synapses, number of optical operands and energy efficiency) with [11] and [29] etc. should be provided to demonstrate the advance of this work.

We have also provided the parameters requested by the referee in Supplementary Note 13-15 for readers' reference. Even though this work is comparable to Refs. [11] and [29] in several aspects, the analogy should not be taken too far: as discussed above, our work studies the optical-neural-network energy efficiency subject to detection noise limit, which is quite different from the prior works. Therefore, the advancement of this work does not lie in scoring higher in engineering metrics when compared to the prior works.

Changes to Supplementary Materials:

- We have supplied the whole system energy consumption and efficiency in Supplementary Note 15.

For readers interested in comparing the energy efficiency of different schemes, we also refer them to the supplementary materials of Ref. [29], which has done an excellent job summarizing the whole-system energy efficiency of many existing schemes.

4. Some specific calculations/numbers of the number of photons per multiplication (such as “0.03, 0.16, 0.32, 0.64, and 3.2 photons per scalar multiplication”) should be presented clearly.

Changes to Supplementary Materials:

- We have added the details of the measurement and calculation in Supplementary Note 15.

Last but not least, we would like to thank the referee again for the constructive comments.

Referee #2:

The paper by Wang et al. presents an original optical computing work, which focuses on demonstrating machine learning tasks with the lowest possible number of photons per Ops. The experiment is done in a free space configuration, using a clever combination of a low-cost incoherent source (a mobile phone screen), optical relays and SLM, and either single detector or a sCMOS camera for more flexibility. This allows to demonstrate a relatively original configuration of optical computing where large dot products are accessible. The paper is, overall, rigorous and well written, with many supplementaries allowing to get an excellent level of details. The discussion is also well done and balanced.

We first would like to thank the referee for thoughtful comments and appreciation of our work. On a high level, the message we tried to convey through this work is a scientific one: ultra-low-energy optical neural networks approaching the photon noise limit can be constructed under very practical experimental settings, potentially across different platforms and schemes. We did not intend to advertise our experimental setup as a complete and practical device that has already achieved competitive performance. Instead, we intended to justify why our choice of setup architecture was the most suitable one for our demonstration experiment. However, we realized the message was not successfully delivered to the readers in the original version of the paper, and we have made several major changes in this revised version. In this spirit, we will try to address the major concerns point-by-point.

R2.a & b

Although the work is solid, I cannot recommend it for publication in Nature, because, in a nutshell, I believe (a) the key result (inference with less than 1 detected photon per ops) is by itself (i.e. proven independently of all the other metrics) not sufficiently novel or surprising, and (b) because I have several fundamental issues about the architecture chosen by the authors to demonstrate it.

R2.a

Regarding the low photon budget per ops, the results are indeed very rigorously performed and I have no reason to doubt them (except a few minor questions and comments, see below). However, it is well known that in machine learning only a low precision is often sufficient, and GPU are nowadays often operated with a low number of bit (in low precision) to save time and memory. It is therefore not particularly surprising that a low SNR (in the few percents as demonstrated here, corresponding to a few bits precision), as a result of high shot noise due to a low photon budget, would result in still decent classification results. Although this regime was not specifically addressed in previous optical computing experiments, I have no doubt they would perform similarly nor am I convinced the current approach is the optimal way to minimise this number. Minimising this figure of merit by simply shortening the integration time, as demonstrated here, irrespectively of all other metrics such as speed, overall consumption, size, etc proves the point, but is in my opinion not really groundbreaking.

R2.a

First, we would like to thank the referee for pointing out an important insight that neural networks can be intrinsically tolerant to low numerical precision, which forms a nice synergy with low-power optical computing. In the revised manuscript, we pointed out this has partially contributed to the possibility of demonstrating the sub-photon multiplication under a very practical experiment setting. Also, we want to point out that not every neural network is equally robust to noise. There are many ways to improve their resilience to noise in physical systems. In this work, we used quantization-aware training (QAT) in combination with several other standard training techniques to train a neural network to work with low SNR, which is described in detail in methods and Supplementary Note 13.

If we understand the criticism correctly, the referee did not find our results novel or surprising because we only showed ultra-low optical energy consumption for an optical neural network without achieving competitive performance in other engineering metrics, such as data throughput rate, size, and overall energy efficiency. We agree that these metrics are important for a competitive engineering prototype, but we do not agree it is necessary to optimize all these metrics to experimentally validate that optical neural networks can operate near the detected photon shot noise limit.

First, we believe our results are novel and of scientific significance to the general field of optical computing: our study is the first experimental investigation on the performance of optical neural networks subject to the detected photon shot noise. Even though several previous studies reported impressive whole-system energy efficiency in terms of J/operation, they did not report if it was possible to operate optical neural networks close to the detected photon shot noise limit. Not only did we present a systematic study on the effects of noise on optical computation, we have shown a somewhat surprising result that optical neural networks can even function with <1 average detected photon per multiplication, which the referee also found intriguing based on the comments below. Though predicted theoretically, there is no guarantee that such a performance can be readily achieved in the real world. We demonstrated that it could be realized under a very practical experimental setting (DIY as referred to by the referee), which is also somewhat surprising. The referee also raised a question on whether our approach was the most optimal one to achieve this performance, and we reason that it does not have to be the optimal way to support the conclusion of this study; in fact, it is probably a good message for other optical-neural-network platforms since they can leverage more advanced engineering capacities to achieve even better results than in this study.

From an engineering perspective, a few instruments used in the experiment could be upgraded or further optimized. For example, the update rate of the OLED display was limited to 60 Hz, which did not constitute a competitive engineering solution for a high data throughput rate. However, the equipment was sufficient to support the validity of the optical energy measurement in this study: the low optical energy demonstrated for optical neural networks was based on photon detection. Since the detector we used had a high bandwidth (~ 3 MHz and a minimum of 100 ns integration time), our conclusion can potentially be extended to systems with much faster input data rates at \sim MHz. The method of controlling detector integration time still proves the

point that the readout of the computational results can be finished within a short time interval, regardless of how fast the input changes.

Changes:

- We mention in the texts how the noise resilience of neural networks can help them to achieve high energy efficiency.

R2.b

On the experimental side, I have no problem at all with the rather DIY and proof of principle demonstration (using a mobile phone screen and non optimised detection system), I find it very ingenious and I am impressed by the feat of aligning 0.5M pixels together. I find however the concept of the approach to vector matrix multiplication by pixel to pixel alignment between a large array of pixel of an OLED panel and an SLM very underwhelming, since contrarily to the claim of the authors, it certainly does not take full advantage of free space (line 312). It does leverage the high number of modes allowed in 2D transverse plane, but completely forget about the ability of free space for interconnects (except for the fan-in and out part, which is hardly treated in the paper). This property of free space, that is exploited naturally in Fourier transforms, convolution, etc, is here totally absent. This results in the fact that the vector matrix multiplication are here done one vector-vector dot product at a time (or few by few is the vector size is lower), sequentially, rather than in parallel as free space optics allows. This in turns results in an extremely slow matrix multiplication process, in stark contrast with for instance other approaches based on 2D convolution. More generally, it does not really exploit the ability of free space optics to perform single shot matrix multiplication rather than dot products as demonstrated here. The resulting speed trade-off is huge (10^4 - 10^5 slower probably), with a probably comparable hit in terms of DAC+ADC energy consumption, and I really don't see this architecture becoming useful in optical machine learning.

R2.b

We agree with the referee that a competitive free-space optical neural network would most likely leverage the all-to-all connection topology offered by optical diffraction in free space. In fact, if the experimental setup is fully parallelized into a matrix-vector multiplier as described in Supplementary Note 10, it would be able to take full advantage of optical parallelism in the sense that each element of the input vector is encoded in light once, and each element in the output vector is read out only once by a detector unit. Even in this case (and in the 4f convolution case mentioned by the referee), a 2D plane can encode arbitrary matrices large enough for most machine-learning applications. A 3D weight encoding scheme (such as in the case of diffractive neural networks or thick random medium) is only necessary when even larger matrices and vector sizes are required.

In this work, the experimental setup serves as scientific apparatus rather than a showcase for a radically new concept for implementing optical matrix-vector multiplication in production. Even though our setup only performs one optical vector-vector dot product at a time, it still proves the point of optical energy efficiency of optical neural networks: a quite commonly used method to implement matrix-vector multiplier is to execute its constituent dot products in parallel, but the

result of each dot product should still be the same as when they were computed separately in sequence (if not, the cause is likely cross-talk due to engineering since these dot products are, by definition, parallel and independent.). More broadly, many other schemes, including the 4f convolution mentioned by the referee, are composed of many parallel executions of optical dot products, even though they are usually not explicitly isolated and called out. The fact that we singled out each dot product potentially enabled our results to be generalized more easily to schemes beyond just matrix-vector multiplication.

From a retrospective point of view, it may be possible to implement a setup that takes full advantage of free-space optical interconnects and meanwhile demonstrates the sub-photon-per-multiplication result in matrix-vector multiplication. However, combining these two messages in a single paper may be misleading since it seems to suggest the particular architecture choice is instrumental for demonstrating the optical energy efficiency. Therefore, the setup was designed to perform the elementary operation of optical dot products well and clean enough in order to study the noise resilience of optical neural networks. In this revision, we decide to further de-emphasize the setup design features that are non-essential for supporting our scientific conclusions.

R2.2

The speed issue is tackled in a rather narrow way and quite exaggerated, based mostly the detector speed (the claim is 10MHz with the ability to go to 10GHz), discarding the SLM and source speed, and the number of frames requires. I doubt whether the number of Ops/second can reach any competitive number with this approach (despite section 15 of the supplementaries which is based mostly on numbers from integrated optics solutions).

R2.2

Originally, we intended to explain the speed of our detector allows our conclusions to be generalized to systems with even faster input data throughput rates. The discussion on why the speed issue does not affect our conclusion has been addressed in more detail above (link). However, we agree with the referee that there are major challenges that need to be overcome for a setup like the one we used in our experiments to achieve speeds anywhere near 10 MHz, given the current limits of many-pixel arrays of light sources and detectors. To avoid the reader getting the impression that we are suggesting an easy path to a practical high-speed implementation using the exact system architecture we proposed, we have removed the discussions about speed, since they are not central to our message about energy efficiency.

Changes:

- We have removed the aforementioned claims.

Changes to supplementary materials:

- We have removed the original Supplementary Note 15 on optical neural network energy scaling since the calculation was based on engineering parameters in the literature, which is not closely related to this work.

- We added a new Supplementary Note 15 that details both optical and whole-system energy efficiency of our setup. Even though it is not the focus of this study, we supplied the information for more transparency on experimental conditions.

For these two main reasons, I believe the broad interest of the work is too limited, and I cannot recommend publication in Nature. The work is nonetheless interesting and original and definitely deserve publication in a good journal, I would for instance certainly support publication much more heartily in e.g. Nature Communications.

Thank you for the recommendation!

See below some comments for the authors to consider:

R2.3

Q: about the discussion around line 167 : I have almost a philosophical question : in incoherent imaging, the contributions adds up in intensity, and there are no interference between different pixels. Therefore, for a large N, there was much less than 1 photon per product, meaning many pixels (OLED+SLM) had no photon at all. This is contrast with coherent techniques where the detected intensity is the result of a global interference between all the path/pixels. Can the authors comments and maybe discuss how the low-photon results would compare in a coherent illumination scheme?

R2.3

We think the question raised by the referee is a good one that may be asked by many readers. Therefore, as mentioned in change 3a, we have added a new paragraph in the discussion session to interpret the sub-photon-per-multiplication result: “The sub-photon-per-multiplication result in this study can be interpreted in the following way: in a single pass of light through the experimental setup for computing element-wise products (i.e., past the modulator in Figure 1b, but before optical fan-in), the number of measured photons in each spatial mode would be an integer if each spatial mode were measured separately. However, after optical fan-in, only the total number of photons across all the spatial modes is measured. Since many of the spatial modes are likely to contribute zero measured photon when they have low expected photon number, the number of photons averaged across all the spatial modes, each performing one scalar multiplication in our ONN, can be between 0 and 1.”

When coherent light is used for experiments, the amplitude instead of the intensity of the spatial modes are summed. Since coherent light contains extra phase information, destructive inference can take place among spatially overlapped optical modes. Such a mechanism can be utilized to represent signed numbers as shown in Ref. [27] in a more regular Stanford Matrix-vector Multiplier. From the perspective of photon detection, the total intensity now also depends on the phase noise of each constituent spatial modes besides the intensity plus photon detection noise of each spatial mode. Despite this technical detail, as long as the noise of each mode is statistically independent of each other to some extent, the aggregation of a large number of spatial modes at the detector would still bring about the benefit of noise reduction.

R2.4

Q: Line 196 : 0.001 corresponds to 500 photons for 0.5M pixels, and I thus find the 6% RMS surprising. Can one estimate the theoretical RMS error assuming shot noise on the same data? So shouldn't the RMS depends on the complexity of the two vectors being multiplied ?

R2.4

The plot below shows how the dot product error depends on photon budget *and vector complexity*. In short, at a low photon budget, the error is dominated by photon shot noise and decreases with photon budget (compare blue and green groups, which were taken from the same test vectors only with a different detector integration time). However, within each group, the error increases with the density of the vectors (number of non-zero elements), which can be regarded as a measure of the “complexity” of the vectors. Overall, the (absolute or normalized) error of dot products on different vectors has a reasonably small spread around the average (e.g., when the average is 5%, the variation is from 1.5~8%). The 6% RMS error is not surprising if one estimates with the SNR of photon shot noise:

$$\begin{aligned}
 \text{SNR} &= 1/\sqrt{\text{total photon counts per dot product}} \\
 &= 1/\sqrt{(0.001 \text{ photons per multiplication}) \times (500\text{k multiplications per dot product})} \\
 &= 1/\sqrt{500} \sim 4.5\%
 \end{aligned}$$

R2.5

For instance for the MNIST results, considering the total number of detected photons as an evaluation of the number of photons per MAC is misleading, as the input images are mostly

black (close to 90%) therefore all photons should be concentrated on a small number of multiplications, at least on the first matrix multiplication between the first two layers. By this I mean that the photon budget results are strongly dataset (and probably network structure) dependent.

R2.5

The referee is correct that MNIST images are sparser than many other datasets, and the per-operation photon budget may vary for different datasets. The less-than-one-photon-per-multiplication result in this work follows closely the simulation results shown in this work and Ref. [7]. Also in Ref. [7], there is simulation indicating the photon budget increases to ~ 10 photons per multiplication when more complex, brighter, and denser images are used as inputs, such as those from ImageNet. This simulation result, to some extent, verifies the speculation of the referee.

Besides datasets, it is also true that the neural network structure and size would also affect the average number of photons per multiplication since wider layers generally allow more redundancy and sparsity. Exactly for this reason, we also note that it is not meaningful just to pursue the lowest possible energy *per operation*. This is because, with wider layers, the average optical energy per operation may decrease, but the *total* energy consumed to classify an image may increase since the total number of operations increases. For this reason, in the revised manuscript, we also include a second metric of detected optical energy *per inference* for a given benchmark task, which is classifying MNIST handwritten digits in this paper. The per-inference energy consumption is a widely used metric in machine-learning hardware community [36, 40, 41]. Even though we have shown some surprising results on less-than-one-photon-per-multiplication with optical neural networks, the true significance lies in the fact that the demonstration was made on a rather standard structure (multi-layer perceptron) of moderate size, according to the convention of the machine-learning community. The neural network comprises 4 layers with widths 784, 100, 100, and 10, respectively, and the total number of multiplications is 90,384 for each forward propagation, which is quite small compared to other similar models achieving similar performance (for examples of models and their classification accuracies, refer to: yann.lecun.com/exdb/mnist/). Given how standard the hardware and software methods employed in this study are, it is reasonable to expect similar results to be generalized to different hardware platforms, as well as different neural-network architectures and datasets.

R2.6

Along the same line, I wonder if some of the training could be done with photon budget in mind, for instance trying to maximise accuracy for a given end photon budget ?

R2.6

Generally, for lower signal-to-noise ratio, the quantization-aware training (QAT) algorithm can be tuned to adapt to a lower bit depth.

R2.7

-about the fan in/out procedure, although how do the author would see a way to do it in a reconfigurable way ? I would see this as a crucial (and very tricky) feature for future use case where matrix multiplication should be of variable size, and I honestly don't see a scalable and easy reconfigurable way to do it in free space (except maybe with multiple SLM and relays), nor in integrated optics.

R2.7

As the referee has pointed out, optical fan-in and fan-out can be made reconfigurable with additional SLMs, but such active elements add to the total energy consumption. The fixed vector size is not a unique problem faced by optical neural networks though. GPUs and TPUs also only work best with certain vector sizes, and the utility rate of their threads decreases when the vector size is not perfectly matched to the number of parallel threads [35].

R2.8

-line 286 : “the experimental result agree *well* ” is maybe an overstatement; I would say there is a good qualitative agreement (the trend is the same in the transition from high to low photon / optics) but it seems that the experiment is noticeably better than the simulations in the intermediary regime, which is surprising and could be certainly discussed further (maybe a calibration issue?).

R2.8

We agree that the agreement between simulation and experiment is reasonable but not perfect. We suspect several possible reasons for the deviation. 1) While it is possible for the calibration to contribute to a certain amount of error (e.g., when the integration time is getting close to the bandwidth of the detector, the detector transients should be considered and the noise equivalent power also increases), it should not account for more than a few percent of errors in optical energy measurement. 2) As seen in Figure 3 and Fig. 3 in Ref. [7], the classification accuracy of an optical neural network is very sensitive around a critical per-operation photon budget, and therefore the exact matching between experiment and simulation can be challenging. The simulation curve in Figure 3b only models photon shot noise but not other instrumental noises, nor does the simulation contain any parameter that can be fitted to experimental results. Taking these factors into consideration, we consider the match between the experiment and simulation to be reasonable.

R2.9

-about the state of the art (paragraph line 76) the vector size on silicon photonics implementation is indeed limited to a few tens at most, but for free space much larger sizes have been reported, even very early for radar imaging in the 70's. More recently, 10^4 - 10^5 are typical and growing rapidly (see within refs 23-29 for instance), so the sheer number of nodes is high but by no means significantly beyond the state of the art (again, with no possibility of large scale vector

matrix multiplication except sequentially). I therefore believe the statement « While the potential for large-scale operation exists based on the available parallelism in free-space spatial modes, this potential has not yet been realized. » (L86) is incorrect. It is not yet fully realised and certainly can still scale up, but is definitely not restricted to a few tens of modes anymore.

R2.9

Changes:

- We have removed the quoted statement and other related statements from the paper since they are controversial and unnecessary for the delivery of the main message of this paper.

R2.10

-line 135 : I would specify that the fan-out is digital/logical (in this incoherent scheme). it becomes clear later but was very confusing to me here.

R2.10

To avoid causing confusion to readers, we now describe our setup as an optical vector-vector dot product multiplier that does not contain either an optical or digital fan-out operation, since the fan-out operation is not strictly needed to prove the point of this study. In Supplementary Note 10, we discussed how optical fan-out would allow parallel processing and expand our setup into a modified Stanford Matrix-vector Multiplier, if it were to be used as a practical device.

R2.11

-I would say a few words in the supplementaries about the current speed of the method, total duration of the experiments (for instance the refresh rate and the number of iterations requires to perform a single inference step for the 4-layer network). I understand they are done at low speed here and it's definitely not the key message, but it would still be interesting.

R2.11

Changes to Supplementary Materials:

- We have supplied computational speed and the whole system energy consumption/efficiency in Supplementary Note 15.

Last but not least, we would like to thank the referee for several insightful questions that allowed us to better convey the scientific results in this manuscript.

Referee #3:

In this paper, the authors propose an experimental study of a proof of principle in optical neural network (ONN) implementation to demonstrate “fundamental energy advantage over electronic neural network implementation”. The authors’ claim relies on a fully detailed experiments showing that a very standard architecture, such as a multi-layer perceptron with 4 layers, can operate in a photon-budget regime which is limited by optical shot noise. To be direct, let me say that these results are interesting regarding precision matrix-vector multiplication in optical domain, but not at all surprising concerning the ONN implementation. Even if the efficiency in terms of low-power operations provided by custom free-space optical matrix-vector multiplier is nicely demonstrated in the proposed work, the main results describes a 2D blocks architecture to perform element-wise modulation of spatial modes after a fan-out provided by an OLED display. The fan-in is done by spherical lens on detectors.

We appreciate the referee’s comments and interest in our optical-dot-product precision results. The referee’s criticism of the optical setup is reasonable. In the revised manuscript, we explain more clearly that the focus of this study is to investigate how the precision of optical dot products and the classification accuracy of ONNs change near the fundamental limit of photon detection shot noise. As a result, we explain why our choice of architecture is suitable for this central scientific message, which is also described in major change 1. We also de-emphasized some of the technical points that are non-essential for supporting the main scientific message. In other words, the optical implementation shown in this work should be viewed as scientific apparatus to enable this study rather than an engineering prototype aimed for competitive performance in all possible metrics. More broadly, the intention of this study is to demonstrate the potential energy efficiency of optical neural networks in a way that is more generalizable to different architectures and platforms, instead of claiming our specific setup design is critical for achieving the demonstrated energy efficiency. Therefore, the fact that the implementation is quite “standard” makes the generalization more straightforward. The significance of the work lies in the demonstrated optical energy efficiency itself, which is novel and of reference value to the broader community of optical neuromorphic computing.

R3.1

The play of optics in the entire neural networks remains limited in the proposed approach, where implementation and training are performed on digital processor. Only inference is performed on a hybrid digital/optical setup, if we consider data workflow description for running the NN for inference through the two hidden layers (as explained in Part III, page 23 in supplementary information document). This is to my point of view, the main limitation in the proposed results because the energy efficiency is demonstrated here for the execution of 500 thousand scalar multiplications in parallel, the approach remains limited terms of future design of a new generation of ONN, as explained in discussion section.

R3.1

If we understand the referee’s comments correctly, the referee considers our optical setup being used only for machine-learning inference as a major limitation. To our knowledge, almost all

recent ONN demonstrations have targeted the inference phase only, and this is currently the main goal of the field [6]. Of course, training is an important problem to solve, but is not the focus of this field at the moment. We note that the training of some architecture requires an electronic computer to change the optical setup parameters based on the measurement of the setup output; however, the algorithm determining how to change the setup parameters is still executed by digital electronic computers, and therefore the training is still performed by the digital computer instead of the optical setup.

R3.2

Another issue concerns the ONN operating speed that is limited to MHz bandwidth. The possibility to increase the computation speed will need to completely change the experimental setup used, because high-speed SLM and matrix detectors are not available for GHz bandwidth, unlike the authors explain at the end of discussion section (lines 344-345) where ref. 37 and 38 are not really compatible with spatial modulations of light.

R3.2

We agree with the referee that the relatively slow update rate of SLM and detector array for spatial optics is a major limitation for high-speed spatial-domain ONNs. We also thank the referee for pointing out the modulators described in the original Ref. [37, 38] are not yet available for modulating spatial light, and we have removed those references. In this study, even though we used the spatial optical modes for the demonstration of ONNs, our conclusion on ONNs operating under extremely low optical energy can potentially be extended to other platforms, including integrated photonics. This is because the experimental results are based on photon detection, and similar kinds of detectors are used in both free-space and integrated photonic platforms.

R3.3

A minor remark concerns the section 15 in supplementary information document concerning the energy scaling for ONN, even it is very useful to try to determine the total energy consumption of an experimental setup, it seems that the study is too far away from the proposed optical free-space setup used to obtain the results presented in Fig 2. and Fig. 3.

R3.3

Changes to Supplementary Materials:

- We have removed the original Supplementary Note 15 on ONN energy scaling since the calculation was based on engineering parameters in the literature, which is not closely relevant to this paper.
- We added a new Supplementary Note 15 that details both optical and whole-system energy consumption/efficiency of our setup. Even though it is not the focus of this study, we supplied the information as experimental conditions.

As a consequence, I do not recommend the publication in Nature journal since it does not

provide a crucial step in ONN implementation, and I do not consider that the proposed results are determinant for the emergence of new generation of optical processor. The results of the manuscript are an improvement concerning the domain of large-scale optical matrix-vector multiplier in free-space optics, but do not really constitute an unexpected new advance of immediate interest to many people in deep learning.

Last but not least, we would like to thank the referee for all the constructive comments which helped us to improve the paper. We would like to add that our experimental results constitute the first systematic experimental investigation on ONNs operating in a regime governed by detected photon shot noise. Our demonstration is significant in the sense that it provides experimental evidence that ONNs working at the physical noise limit can be realized in the real world. For this reason, our work should be of interest to the broader community of ONNs across different engineering platforms (free-space or integrated photonics) and employing different multiplexing methods (e.g., spatial, temporal, and wavelength multiplexing).